# Dynaboard: An Evaluation-As-A-Service Platform for Holistic Next-Generation Benchmarking

**Zhiyi Ma**[†*]   **Kawin Ethayarajh**[‡*]   **Tristan Thrush**[†*]   **Somya Jain**[†]

**Ledell Wu**[†]   **Robin Jia**[†]   **Christopher Potts**[‡]   **Adina Williams**[†]   **Douwe Kiela**[†]
[†] Facebook AI; [‡] Stanford University
dynabench@fb.com

## Abstract

We introduce Dynaboard, an evaluation-as-a-service framework for hosting benchmarks and conducting holistic model comparison, integrated with the Dynabench platform. Our platform evaluates NLP models directly instead of relying on self-reported metrics or predictions on a single dataset. Under this paradigm, models are submitted to be evaluated in the cloud, circumventing the issues of reproducibility, accessibility, and backwards compatibility that often hinder benchmarking in NLP. This allows users to interact with uploaded models in real time to assess their quality, and permits the collection of additional metrics such as memory use, throughput, and robustness, which – despite their importance to practitioners – have traditionally been absent from leaderboards. On each task, models are ranked according to the Dynascore, a novel utility-based aggregation of these statistics, which users can customize to better reflect their preferences, placing more/less weight on a particular axis of evaluation or dataset. As state-of-the-art NLP models push the limits of traditional benchmarks, Dynaboard offers a standardized solution for a more diverse and comprehensive evaluation of model quality.

## 1   Introduction

Benchmarks have been critical to driving progress in AI: they provide a standard by which models are measured, they support direct comparisons of different proposals, and they provide clear-cut goals for the research community. This has led to an outpouring of new benchmarks designed not only to evaluate models on new tasks, but also to address weaknesses in existing models [49, 55, 34], and expose artifacts in existing benchmarks [42, 19, 25, 33, 22, 37]. These efforts are helping to provide us with a more realistic picture of how much progress the field has made.

To date, the metrics by which we assess system performance have received much less systematic attention. Even as the benchmarks have changed, the community has continued to rely heavily on accuracy as the sole primary metric. This gives rise to a "leaderboard culture" in which accuracy is the only thing that matters, even in contexts in which other pressures – e.g., compactness, fairness, efficiency – are clearly important [60, 6, 16, 63, 68, 38, 2, 51, 17]. Recently, Ethayarajh and Jurafsky [17] provided a microeconomic framing of the problem: leaderboard viewers are *consumers* of models, and each viewer has their own set of preferences: some care only about accuracy, others value compactness in addition to accuracy, and so on. Static leaderboards that only rank by model performance thus have a scoring function that is misaligned with the preferences of most users. Moreover, merely focusing on a single metric, such as accuracy, limits the scope of possible issues

---

[*]Equal contribution.

even to consider. In other words, focusing on one standard metric, like accuracy, lets researchers off the hook too much on other pressing issues.

Reproducibility, accessibility, and backwards compatibility are important issues as well. There is a reliance on self-reported results with no trust guarantees [75], and many state-of-the-art claims are famously difficult to reproduce [52]. In some cases, the state-of-the-art model cannot be accessed without sufficient computational resources and/or technical expertise, and older models are often incomparable to newer ones because they have not been evaluated on the same data.

To begin to address these issues, we propose **Dynaboard**. Dynaboard's evaluation-as-a-service backend allows models to be submitted directly to be evaluated in the cloud, circumventing issues of reproducibility, accessibility, and backwards compatibility. Older models can be compared to newer ones because any model can be evaluated on demand. Dynaboard supports reasoning about many different metrics – not just standard accuracy-style metrics, but also memory use and throughput (i.e., inference speed) on identical hardware, robustness, fairness, and so forth. Building on [17], we borrow from microeconomic theory to aggregate these metrics – along with performance – into a *Dynascore* that is used to rank models. The frontend of Dynaboard is a dynamic leaderboard that allows users to customize the Dynascore by placing more/less weight on a particular metric or dataset. As the user modifies the weights that govern the Dynascore, models are re-ranked in real time. Users can also directly interact with a model via the platform, receiving real-time model predictions that help them understand a model's capabilities and limitations, and allowing the community to find challenging examples where the current state of the art still falls short. Together, this allows users to develop and directly express the complex and interrelated values they have for their models.

Although other platforms – CodaLab, DAWNBench [10], and ExplainaBoard [39], to name a few – address some subset of the problems we describe above, Dynaboard is the first to address all of them in a single end-to-end system. It requires minimal overhead for model creators wishing to submit their model for evaluation, but offers maximal flexibility for users wishing to make fine-grained comparisons between models. As state-of-the-art NLP models push the limits of traditional benchmarks, Dynaboard offers a standardized solution for creating the next generation of benchmarks in a manner that allows for the diverse and comprehensive evaluation of model quality.

## 2   Objectives

Dynaboard aims to address the following issues in our current model evaluation paradigm:

**Reproducibility**   The reliance on self-reported results with no trust guarantees [75] makes many state-of-the-art claims difficult to reproduce [52]. Even the choice of random seed can lead to substantially different results [15], and improvements on the state-of-the-art are often not statistically significant [8]. Implementational differences in evaluation metrics can also lead to different scores [54]. Given that the test data for static benchmarks is often publicly available, reported results may also be the outcome of overfitting hyperparameters [4].

**Accessibility**   Whichever model happens to hold the "state-of-the-art" title should be accessible by as many researchers as possible. Democratizing model evaluation is essential to understanding our current weaknesses, for making progress in the long tails of the data distribution, and for collecting new adversarial datasets where the state-of-the-art fails. Just open sourcing the model weights is insufficient, because even when given trained models, inference-time compute resources may not be readily available in many parts of the world.

**Backwards Compatibility**   Dynamic human-in-the-loop data collection is a great alternative to static benchmarks, but as new evaluation sets are introduced in later rounds, old models become incomparable to new ones, simply because they haven't been evaluated on the same data. It should be possible to evaluate *a model* on demand, rather than the model's *predictions* at a single point in time.

**Forwards Compatibility**   Automated metrics such as BLEU and ROUGE have many known flaws, and creating better automated metrics (e.g., BLEURT [64], BERTScore [77]) is an active area of research. However, the current paradigm does not allow for old models to be evaluated on new automated metrics that come later, since it relies on reporting from the model creator. If our field comes up with new and better metrics, we should be able to immediately gauge overall model quality.

**Prediction Costs**   Most leaderboards treat the cost of making predictions as zero, which does not hold in practice; memory use, latency, throughput, and lack of robustness – among many other factors – have real-world implications in deployment [60, 17]. A highly accurate model may be useless to an embedded systems engineer if it is untenably large, for example. For users to be able to make informed decisions about the risks and rewards of using a particular model, they need to be given as much information as possible.

**Utility Estimation**   There is no single correct way to rank models: every leaderboard viewer has a different set of preferences, as manifested in their utility function [17]. Some users only care about model performance, but others care about memory use, throughput, fairness, and more. A leaderboard should not be overly prescriptive when ranking models – it should provide a default ranking, but ultimately give users the freedom to customize the scoring/ranking function to better approximate their utility function.

The evaluation-as-a-service backend of Dynaboard addresses the first four of these. By having models submitted directly to the platform, they can be accessed and evaluated on demand on any dataset (see Section 4). The frontend of Dynaboard is a dynamic leaderboard that addresses the last two issues: prediction costs and utility estimation. Although the leaderboard has a default ranking, the scoring function is constructed in a principled way, borrowing from economic theory to estimate the rates at which users are willing to trade-off prediction costs for performance. More importantly, users are given the freedom to manipulate the scoring function to better approximate their preferences.

## 3   Related Work

**Evaluation platforms**   Although other platforms have addressed some subset of the issues in Section 2, they have not addressed all of them in a single end-to-end solution. DAWNBench [10] reports some prediction costs, such as the time needed to reach a particular accuracy. However, it still relies on self-reported results, meaning that reproducibility, accessibility, and forwards/backwards compatibility are still issues. CodaLab[2] and EvalAI[3] make it significantly easier to access models and reproduce stated claims, but automatically evaluating any model on any new dataset or metric is not straightforward, making backwards/forwards compatibility hard to scale. It would also be difficult to collect prediction costs on standard hardware. Explainaboard [39] and Robustness Gym [23] allow for a more nuanced comparison of models, but through the lens of which slices of data a model performs well on and what types of errors it makes. Models are not directly uploaded, however, so prediction costs are not collected and forwards compatibility cannot be automated. We see these approaches as complementary to Dynaboard, helping provide a richer view of model performance.

**Multi-metric evaluation**   There have been some attempts to rank models by metrics other than accuracy. Although not explicitly framed in terms of utility, Mieno et al. studied the trade-off users were willing to make between accuracy and latency in speech translation [44]. Some challenge tasks in recent years have required the reporting of prediction costs or have instituted limits on the maximum costs that can be incurred, as in the EfficientQA challenge [27, 45]. However, such practices are still few and far between. Our hope is that over time Dynaboard can help lower the overhead required to create multi-metric benchmarks and help them proliferate and diversify, such that every sub-community can have its own canonical benchmarks, like GEM [21] for natural language generation, BIG-Bench[4] for language model probing, or GLUE [73] and SuperGLUE [72] for NLU.

## 4   Backend: Evaluation-as-a-Service

The evaluation-as-a-service pipeline uses a standardized application programming interface (API) that is similar to other model evaluation platforms like EvalAI and Codalab that allow creators to upload models. Researchers submit their models for evaluation in our *model evaluation cloud*, in which all models are scored in exactly the same way, on the exact same data, within the same (virtual) hardware constraints. Testing models on different datasets, from standard well-known test sets to

---

[2]https://codalab.org
[3]https://eval.ai
[4]https://github.com/google/BIG-bench

stress tests [47, 11] and check lists [59] can be cumbersome – we offer evaluation-as-a-service (EaaS) for NLP models. We ask model developers to specify model cards [46] to report on and properly document their contributions. In this paradigm, it is no longer possible for people to cherrypick results: the same model instance is evaluated on all evaluation sets.

We take a *multi-metric* approach, going beyond just accuracy, where models are evaluated on multiple axes of evaluation. The top models can be employed in-the-loop for dynamic adversarial evaluation [34], and all uploaded models can be made accessible online so that anyone can "talk to" these models (in a corresponding frontend UI) to see how they perform. Models are evaluated on combinations of adversarial and non-adversarial datasets. All statistics are made available to the user, who can rank models by specifying which datasets and which metrics should factor into the ranking and in what proportion (see Section 5 for details).

Importantly, we do not claim to have built or discovered the one right approach for model evaluation. The goal of Dynaboard – and the Dynabench data collection platform it is built on – is making *dynamic* as many of the components employed by the current status quo as possible: i) rather than having a single metric, we can have many metrics, with the ability to add more as new automated metrics are conceived; ii) rather than having a fixed set of evaluation datasets, we can have many and add more over time; iii) rather than evaluating models only on static datasets, they can be evaluated dynamically in the loop. For this approach to succeed, the ability to re-evaluate older models is essential: when new datasets are introduced, new metrics come out, and as new community preferences emerge, we need to be able to see where we stand with our current state-of-the-art.

## 4.1 Metrics

All models are evaluated along multiple axes using a variety of metrics. The ideal model is not only accurate, but fast, memory-efficient, fair and robust. Many of these properties are associated with trade-offs: a binary classifier that always outputs 1 is very fast and uses little memory, but is very inaccurate. While we might incorporate additional metrics in the future – one of the advantages of our platform is that we are able to do so – the currently reported set of metrics is as follows:

**Performance** The standard evaluation metric in machine learning is some form of performance, whether it is accuracy, F1, AUROC, BLEU, or something else. The exact performance metric, in our case, is task dependent. One task can have multiple performance metrics, but only one metric is used as the canonical performance metric when ranking models. When metrics are aggregated to rank models, our default weighting places half the weight on the canonical performance metric and splits the remaining half among the others (see Section 5 for details).

**Throughput** The amount of inference-time compute required by a given model is defined as the number of examples it can process per second on its instance in our evaluation cloud. For fair comparison, models are deployed on the exact same architecture. Measuring the computational efficiency of a model is important for two main reasons. First, a highly accurate model that takes a very long time to label a single example has limited use in most real-world scenarios. Second, as a field, we need to focus more on *Green AI* and explicitly account for a model's energy/carbon footprint in our model ranking [63, 68, 28].

**Memory** The amount of memory that a model requires is given in gigabytes of memory usage, averaged over the duration that the model is running, with measurements taken over a fixed interval. Note that unlike with the other metrics, the goal is to minimize memory use rather than maximizing it. For this reason, before aggregating all the metrics into a single score for ranking models, we transform "memory used" into "memory saved" by subtracting it from the maximum possible (virtual) memory that can be used (see Section 5 for details).

**Fairness.** There is no single, best way to measure model fairness [35, 50, 3]. Whether to include any explicit fairness metric was a difficult choice – we acknowledge that we might inadvertently facilitate false or spurious fairness claims, fairness-value hacking, or give people a false sense of fairness. The research community has a long way to go in terms of developing well-defined concrete fairness measurements. Ultimately, however, we came to the conclusion that fairness is such an important axis of evaluation that we would rather have an imperfect metric than no metric at all: in our view, a multi-metric evaluation framework simply *must* include fairness as a primary axis.

Table 1: Scoring and evaluation-as-a-service datasets used for each of the four Dynabench tasks. Unless otherwise indicated, scoring sets are the respective test sets.

| Task | Scoring | Evaluation-as-a-Service |
|------|---------|-------------------------|
| NLI | SNLI [5], MNLI [74] matched and mismatched, ANLI rounds 1-3 [49] | Respective dev sets; HANS [43], NLI stress tests [47], Winogender [62] recast as NLI [53] |
| QA | SQuAD dev [57], AdversarialQA [1] (Dynabench QA round 1) | AdversarialQA dev, the 12 dev sets from MRQA shared tasks [18] |
| Sentiment | SST3 [66], DynaSent [55] | Respective dev sets; Amazon Reviews [78] test and dev (10k subsample), Yelp Reviews [78] test and dev (10k subsample) |
| Hate speech | Learning From The Worst [71] | Respective dev sets; HateCheck [61] |

We compute the fairness metric based on demographic parity using a black box evaluation protocol, without assuming any access to the model itself beyond getting its predictions. Thus the question becomes: how do we construct or manipulate input data such that we can assess some aspects of a model's fairness by inspecting its outputs? Following the precedent set by datasets such as Winogender [62] and WinoBias [79], we perform perturbations of original datasets in two ways: (i) by substituting noun phrases that unambiguously encode a gender identity with those that have the same part-of-speech but refer to another identity (e.g., replacing "he" with "they" following [14, 13]) and (ii) by substituting names with others that are statistically predictive [70] of another race or ethnicity (e.g., replacing "James" with "Jamal"). This way, a model is considered more 'fair' if its predictions do not change after perturbation, and considered less fair if they do. More details are in Appendix A. As the community invents better and better black box fairness evaluation metrics, which we hope they will, we can re-evaluate older models on new datasets and new metrics, due to our backwards-compatible evaluation paradigm. Note that while we use fairness as a metric, we can and do also report results on fairness datasets [62] as part of our evaluation-as-a-service platform.

**Robustness.** This is a question of how robust a model is to a particular set of (mostly demographic) axes, and there are many other factors that can be involved in a model's robustness. In order to capture this aspect of model quality, we also conduct black box robustness evaluation in a similar manner to fairness. That is, following past work on NLP robustness [30, 31], we perturb examples and measure whether a model's prediction changes. Specifically, we use the recently released TextFlint[5] evaluation toolkit [24] to do so (details are provided in Appendix B). Initially, we focus mostly on typographical errors and local paraphrases – e.g., a "baaaad restuarant" is not a good restaurant – but just like for fairness, we fully expect this metric to evolve and improve over time.

## 4.2 Tasks

Currently, there are leaderboards for four tasks: Natural Language Inference (NLI), Question Answering (QA), Sentiment Analysis and Hate Speech. We evaluate on a combination of adversarially collected datasets for these four tasks, as well as datasets from other sources. Each task has its own set of task owners who control the default setting, i.e., which datasets are included in the corresponding benchmark and in the evaluation suite, how the datasets are weighted, and how the metrics are weighted relative to each other for that particular task. Note that the leaderboard rank is only determined based on a subset of the full set of evaluation-as-a-service datasets (see Table 1).

# 5 Frontend: Dynamic Leaderboard

The frontend of Dynaboard aggregates the various metrics into a single score for ranking models. Users can manipulate the parameters of this scoring function to better reflect their preferences.

A static leaderboard's ranking cannot approximate the preferences of the leaderboard viewer, because it cannot consider (e.g. computational) costs. Ethayarajh and Jurafsky [17] frame this shortcoming in economic terms: a leaderboard viewer is a *consumer* of models and the benefit they get from a model

---

[5]https://www.textflint.com

is its *utility* to them – each individual has their own utility function. The scoring function implicit in ranking by accuracy or F1 is misaligned with the utility function of most users; for example, an IoT engineer looking to find the fastest and smallest model getting 80% accuracy would be poorly served by a static leaderboard.

In reporting many prediction costs, Dynaboard addresses the first of these problems. But how do we combine these disparate metrics into a single score that can be used to rank models? Additionally, how can we allow Dynaboard users to align the scoring function with their own utility function? We propose a method for calculating "exchange rates" between metrics that can be used to standardize units across metrics, after which a weighted average is taken to get the Dynascore. The user can adjust the default weights – e.g., our IoT engineer can place more emphasis on memory and throughput – to approximate their utility function; as they do so, the models will be dynamically re-ranked.

## 5.1 Background

Each model has $k$ properties that informs its utility – throughput, memory, accuracy, etc. – which can be thought of as a separate *good*. By definition, a good is something one always wants more of, meaning that "memory saved" is a good but "memory used" is not, for example. When a metric is not naturally a good, we transform it into a good by subtracting it from a budget cap or the maximum of that metric across all models. For example, subtracting memory used from a maximum available memory of 16 GB gives us "memory saved", which we want to maximize. This standardization allows us to use the same method when calculating the trade-off between any two metrics.

A model is a point in this space of goods, and a user's utility function maps each point to an amount of utils. An *indifference curve* is a set of points that provide the same utility and can be thought of as a level curve of the utility function [41]. Indifference curves are monotonically negatively-sloped: if one model is strictly better on all dimensions than another, then the two cannot lie on the same negatively sloped line, because the former will have strictly greater utility. For example, for a more accurate model to be on the same curve as a less accurate one, the former may have to use up more memory. For a given indifference curve, the rate at which this trade-off is made – i.e., the negative of its slope at a point – is called the *marginal rate of substitution* (MRS) [41].

## 5.2 Converting between Metrics

To calculate the default Dynascore, we first estimate the rate at which users are willing to trade-off each metric for a one-point gain in performance (i.e., MRS with respect to performance) and use that to convert all metrics into units of performance. Once converted, a weighted average is taken to get the final score. We choose this approach for two reasons. First, by drawing from the microeconomics literature, it is a principled approach to user personalization that builds on the utility-based critique of static leaderboards [17]. Second, typical normalization methods (e.g., $z$-scores) do not work well when a metric's values are highly skewed, as they often are in practice. For example, as seen in Table 2, taking a weighted average of $z$-scores would lead to T5 ranking worse than the majority baseline on many tasks; this happens because T5's high memory use is as much an outlier as the baseline's poor accuracy. Even BERT, a more lightweight model, ranks worse than the majority baseline on NLI and QA when ranking by the average $z$-score, which is highly unintuitive.

Given that each user has their own unique utility function – and thus unique indifference curves – each has a different rate at which they are willing to make a trade-off between a metric `M` and the performance metric `perf`. This is fine; the default "exchange rates" between the metrics should merely be a starting point that can be adjusted by the user as they see fit (see Section 5.3). To calculate the default exchange rates between some metric `M` and `perf`, we consider the simplest possible case, where each indifference curve only lies in two dimensions – i.e., the utility changes only with respect to `M` and `perf` – and all models lie on the same indifference curve. This assumption is needed because we do not have direct access to any $k$-dimensional utility function; our data is sparse.

We then calculate the *average marginal rate of substitution* (AMRS) of this indifference curve, which tells us the rate at which model creators, as a group, are trading off `M` for a one-point increase in `perf` while keeping utility constant. For example, if AMRS of "memory saved" with respect to accuracy were 0.5 GiB, then each GiB of memory saved would on average be worth 2 points of accuracy. By dividing `M` by **AMRS**(`M`, `perf`), we can convert it to units of performance.

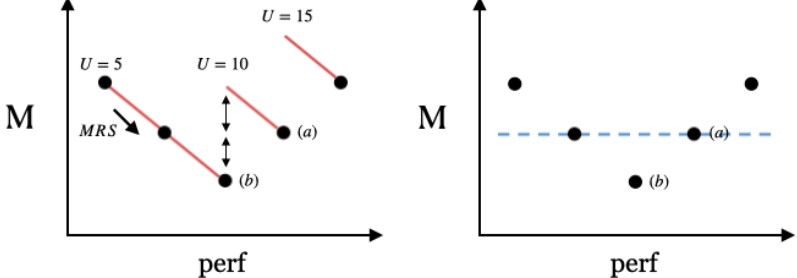

Figure 1: Left: The marginal rate of substitution (MRS) is the negative of the slope of an indifference curve (shown in red) with fixed utility $U$, each of which shows the trade-off being made between metric M and performance perf. Since model (a) and (b) cannot possibly lie on the same curve – because the former is better in both respects – we assume that when all else (including utility) is held constant, the increase in performance from (b) to (a) should come at the *expense* of the increase in M, giving us an estimate of where the higher indifference curve lies. Right: Taking the line-of-best-fit to estimate this trade-off would not work when some models are strictly better than others.

Note that this means that the Dynascore is inherently dynamic and incorporates the AMRS *at a given point in time*, and so is subject to change as more models are added.

**Assumptions**    What if there were no trade-off between a metric and performance? For example, performance might rise with robustness up to a point and come at the cost of robustness past that. If the $i$th most accurate model $x_i$ had both better performance and were more robust than the next most accurate model $x_{i+1}$, then they could not possibly lie on the same indifference curve, since having more of both goods is axiomatically preferable to having less of each. To tackle this problem, we make two assumptions: (i) if $M(x_i) > M(x_{i+1})$ and $perf(x_i) > perf(x_{i+1})$, then models $\{x_j | j > i\}$ lie on a lower indifference curve than $x_i$, though not all necessarily lie on the same one; (ii) there exists a model $\langle perf(x_{i+1}), M(x_i) + (M(x_i) - M(x_{i+1})) \rangle$ on the same indifference curve as $x_i$ (i.e., $x_i$ and $x_{i+1}$ would be on the same curve if the latter obtained $M(x_i) - M(x_{i+1})$ more on metric M). In other words, all else (including utility) held constant, we assume that the increase in performance from $x_{i+1}$ to $x_i$ should come at the *expense* of the change in M. This is just an estimate, but it provides a good starting point from which users can choose their own AMRS (Section 5.3).

These assumptions, visualized in Figure 1, allow every model to be represented in the AMRS calculation. Under these assumptions about where the higher indifference curve lies, calculating the MRS is equivalent to taking the absolute value of the slope between points (rather than just taking the negative of the slope)[6].

$$\mathbf{MRS}(M, perf) = \left\{ \left| \frac{M(x_i) - M(x_{i+1})}{perf(x_i) - perf(x_{i+1})} \right| \mid 1 \leq i < n \right\}$$

$$\mathbf{AMRS}(M, perf) = \overline{\mathbf{MRS}}$$

(1)

### 5.3   Weights and Customization

After converting all the metrics into units of performance, we take a weighted average of the converted values to get the Dynascore for a model. For a model $x_i$ with normalized weights $w_M \in \mathbb{Z}$:

$$\texttt{Dynascore}(x_i) \triangleq \sum_M w_M \cdot \frac{M(x_i)}{\mathbf{AMRS}(M, perf)}$$

(2)

The default normalized weights are $w_{perf} = 0.5$ and $w_{M \neq perf} = 0.5/(m-1)$. In other words, performance makes up half the Dynascore while the remaining half is split equally among the other metrics. This is because, in theory, gains in all the other metrics may come at the expense

---

[6]Since the MRS is undefined when $perf(x_i) - perf(x_{i+1})$ is zero, we ignore increases in performance that fall below a predetermined threshold epsilon, or are $\epsilon$-small. We currently set $\epsilon$ to $1 \times 10^{-4}$ for all tasks.

Table 2: Non-dynamic leaderboards for the Dynabench tasks, sorted by the Dynascore. Performance (Perf.) is measured by F1. Note that ranking by the average $z$-score (with the same metric and dataset weighting as the Dynascore) leads to a far more unintuitive ranking than with Dynascore – e.g., T5 often ranks worse than the majority baseline because its memory use is such an outlier. Also note that although "memory used" is reported, before incorporating it into the Dynascore and $z$-score, we subtract it from the maximum possible 16GB to get "memory saved" to be consistent and express it as a utility (i.e., having something we want to maximize).

| Task | Model | Perf. | Throughput | Memory | Fairness | Robustness | Dynascore | Avg $z$-score |
|---|---|---|---|---|---|---|---|---|
| NLI | DeBERTa | 69.54 | 7.41 | 5.71 | 91.97 | 75.70 | 38.83 | 0.24 |
| | RoBERTa | 69.07 | 9.23 | 4.82 | 90.94 | 74.82 | 38.61 | 0.24 |
| | ALBERT | 67.29 | 9.60 | 2.18 | 89.94 | 74.12 | 37.72 | 0.26 |
| | T5 | 67.16 | 7.10 | 10.62 | 91.89 | 73.47 | 37.53 | -0.07 |
| | BERT | 64.82 | 9.39 | 4.13 | 92.11 | 66.38 | 36.36 | 0.06 |
| | Majority Baseline | 32.41 | 77.33 | 1.15 | 100.00 | 100.00 | 22.78 | 0.10 |
| | FastText | 31.29 | 73.94 | 2.20 | 83.23 | 69.14 | 21.13 | -0.83 |
| QA | DeBERTa | 76.25 | 4.47 | 6.97 | 88.33 | 90.06 | 45.92 | 0.48 |
| | ELECTRA-large | 76.07 | 2.37 | 25.30 | 93.13 | 91.64 | 45.79 | 0.33 |
| | RoBERTa | 69.67 | 6.88 | 6.17 | 88.32 | 86.10 | 42.54 | 0.27 |
| | ALBERT | 68.63 | 6.85 | 2.54 | 87.44 | 80.90 | 41.74 | 0.16 |
| | BERT | 57.14 | 6.70 | 5.55 | 91.45 | 80.81 | 36.07 | -0.02 |
| | BiDAF | 53.48 | 10.71 | 3.60 | 80.79 | 77.03 | 33.96 | -0.44 |
| | Unrestricted T5 | 28.80 | 4.51 | 10.69 | 92.32 | 88.41 | 22.18 | -0.52 |
| | Return Context | 5.99 | 89.80 | 1.10 | 95.97 | 91.61 | 15.47 | -0.27 |
| Sentiment | DeBERTa | 76.07 | 7.50 | 4.80 | 94.08 | 79.21 | 71.31 | 0.34 |
| | RoBERTa | 73.74 | 8.95 | 4.14 | 93.87 | 77.81 | 70.11 | 0.28 |
| | T5 | 73.20 | 7.12 | 9.06 | 93.44 | 77.99 | 69.32 | 0.00 |
| | ALBERT | 70.61 | 10.24 | 2.04 | 93.34 | 78.44 | 68.73 | 0.28 |
| | BERT | 68.71 | 8.83 | 6.04 | 93.49 | 72.75 | 66.81 | -0.07 |
| | Majority Baseline | 35.93 | 35.14 | 1.07 | 100.00 | 100.00 | 57.94 | -0.27 |
| | FastText | 53.32 | 32.54 | 1.69 | 78.52 | 65.82 | 57.39 | -0.57 |
| Hate Speech | DeBERTa | 81.34 | 6.20 | 5.40 | 83.58 | 81.94 | 48.31 | 0.23 |
| | RoBERTa | 80.26 | 6.61 | 3.67 | 85.02 | 79.09 | 47.77 | 0.26 |
| | ALBERT | 76.84 | 8.01 | 2.25 | 84.50 | 79.98 | 46.18 | 0.23 |
| | BERT | 76.58 | 7.26 | 3.22 | 86.45 | 77.35 | 45.97 | 0.15 |
| | T5 | 76.59 | 5.48 | 10.47 | 86.71 | 78.52 | 45.80 | -0.19 |
| | Majority Baseline | 54.69 | 16.48 | 1.09 | 100.00 | 100.00 | 37.05 | 0.24 |
| | FastText | 49.70 | 15.34 | 2.61 | 80.09 | 71.12 | 32.46 | -0.93 |

of performance, so giving each metric a weight of $1/m$ would favor degenerate models due to their efficiency and close-to-zero memory. To avoid this, the weighting was split evenly between performance and the (hypothetical) costs of performance.

A Dynaboard user can customize the Dynascore to better approximate their own utility function by adjusting the un-normalized weights. In doing so, they are effectively expressing the average rate at which *they* would be willing to trade off metric M against performance. That is, using the user-specified normalized weights $z_\text{M}$ is equivalent to using the default normalized weights with an effective exchange rate of $(w_\text{M}/z_\text{M})$**AMRS**(M, perf). If a Dynaboard user cares more about a metric M than in our default setting (i.e., $w_\text{M} < z_\text{M}$), then the adjusted AMRS will decline to reflect the fact that they are less willing to sacrifice M for a marginal increase in performance. A possible line of future work is studying NLP practitioners' preferences over time and across different niches (e.g., industry vs. academia). Asking such questions has only been made possible by adopting this utility-based framework for Dynascore.

### 5.4  Limitations & Future Work

In general, the AMRS estimates are better when our assumptions in 5.2 need not apply – i.e., when the models can all lie on the same convex indifference curve. However, we stress that the estimates are only a starting point and are meant to be manipulated by the user by adjusting the weights.

**Identical Models**  Since every metric is converted into units of performance, if *all* models perform exactly the same, the AMRS cannot be calculated. The reason we use performance as the base metric is because every task is guaranteed to have *some* performance metric, while the other metrics may differ substantially across tasks. However, it is possible to pick a different base metric in principle: we could calculate the AMRS of M w.r.t. memory saved, for example, and convert everything into

GiB. Similarly, the converted value for M will be undefined if all models obtain exactly the same value for M – this implies that model creators are not willing to sacrifice M at all for an improvement in performance and therefore that the metric is infinitely more valuable relative to others. Both these issues are unlikely to occur in practice, however.

**Threshold Sensitivity**   We choose to ignore increases in performance less than $\epsilon = 1 \times 10^{-4}$ so that the MRS is never undefined, which we consider reasonable since almost all existing leaderboards only report to 1 or 2 decimal points, so an improvement of $1 \times 10^{-4}$ would be indistinguishable to the user. One could reasonably argue that this threshold needs to be higher however, since many NLP datasets are too small for a 0.01 or even 0.1 increase to be statistically significant [8]. Changing $\epsilon$ can change the Dynascore and the model rankings. However, to avoid potentially confusing the user with adaptive thresholds based on statistical power tests, we chose to fix $\epsilon$ at a conservative value.

**Human Validation**   Not all models and tasks lend themselves to be evaluated automatically; some need humans in-the-loop, either by definition or over time as our models improve. In the future, we hope to incorporate the validated model error rate (vMER) [34] – the number of human-validated model errors divided by the total number of examples – as a metric on the leaderboard, such that any benchmark can use vMER as a base performance metric.

## 5.5   Results

We take models that currently represent the state-of-the-art on our tasks and put them through our evaluation pipeline. Specifically, we finetune and evaluate BERT [12], RoBERTa [40], ALBERT [36], T5 [56] and DeBERTa [26] on all tasks, with the addition of ELECTRA [9] for QA. This set of models roughly encompasses the top 5 models on GLUE [73] and SuperGLUE [72]. As baselines, we add FastText [32] for sentiment and hate speech and BiDAF [65] for QA. We also compare to the majority baseline for the classification tasks (using the training set majority) and for QA a baseline that simply returns the entire context. These baselines are obviously efficient and robust to perturbations, but have low performance. See Appendix C for further details. We compute the Dynascore for each model using the default weights for the metrics and giving equal weight to all datasets. Recall that the Dynascore is dynamic and incorporates the AMRS at a given point in time, and so is subject to change as more models are added.

The results are reported in Table 2. We find that the SuperGLUE ranking is roughly preserved, with Transformer models clearly outperform their predecessors and the baselines. Even when we factor in the additional axes of evaluation, DeBERTa still performs best. In some cases, the gap is actually larger: in terms of accuracy on SuperGLUE, DeBERTa outperforms T5 by 1.1% , but on the sentiment task it's 3.1%, while being more efficient, as reflected in the Dynascore. However, there are some differences with the SuperGLUE rankings, most notably with T5, which ranks much lower here because of its relatively low throughput and high memory use. It is also worth noting that FastText does consistently worse than the majority baseline, even when it is more accurate, since the gain in accuracy is not enough to offset its sensitivity to fairness and robustness perturbations.

The somewhat old-fashioned medium of a paper does not do justice to a dynamic concept like a leaderboard with specifiable utility function, and linking to the actual implementation would break anonymity. We refer the reviewer to the supplementary material for examples.

## 6   Conclusion

We introduced an end-to-end solution for hosting next-generation benchmarks enabling a more holistic and comprehensive evaluation of model quality. Our evaluation-as-a-service platform addresses several important shortcomings in the current status quo, from reproducibility to accessibility to backwards/forwards compatibility. Dynaboard also enables the collection of prediction costs and factors these costs into the overall ranking of models, which users can customize to better align with their own preferences. With NLP models playing an increasingly important role in our daily lives, deciding which model is better than another one is becoming a crucial problem, both for driving further progress and ensuring that we deploy our systems responsibly. We hope that Dynaboard can allow benchmarks to proliferate and diversify, paving the way for the next wave of advances in NLP.

## Acknowledgments and Disclosure of Funding

We thank our collaborators in the Dynabench team, especially Max Bartolo, Yixin Nie and Bertie Vidgen, for helpful comments and suggestions. We're grateful to Amanpreet Singh and Sujit Verma for useful feedback and engineering suggestions. We thank Eric Smith and April Bailey for helping with name lists for the fairness perturbation. We also thank Devi Parikh, Alicia Parrish, Pedro Rodriguez, Dan Jurafsky, Ethan Perez, Kyunghyun Cho and Alex Wang for comments on an earlier draft, and Gargi Ghosh for her support throughout. We thank the TextFlint team for their help on the toolkit. Kawin Ethayarajh was supported by an NSERC PGS-D.

## Broader Impact

**Green AI & Ethical AI**   A key barrier to the adoption of Green AI [63] has been the incentive structure in NLP leaderboard culture, which places emphasis on higher performing (i.e., more accurate) models to the detriment of models that are more lightweight and efficient [17]. Although there are exceptions to this – such as the EfficientQA challenge task [45] – the dominant paradigm still largely ignores these factors. Changing incentive structures is important because they ultimately shape what kind of problems the NLP community works on and what kind of models they build. By drastically lowering the overhead needed to host multi-metric benchmarks, Dynaboard will hopefully allow benchmarks to keep fulfilling their crucial role in driving research progress, while incentivizing the creation of models that are "greener" and more fair, among other qualities.

**Fairness**   Our perturbation-based black box fairness evaluation method is an initial attempt at measuring fairness for NLP tasks/models, and we fully expect this to evolve over time as advances are made in the field. Firstly, our method assumes that a fair system will be one that treats all genders and races/ethnicities equally. We take equal treatment (treating all groups equally) to be a first step towards a longer term goal of equitable treatment (treating all groups fairly in accordance with their needs or circumstances). The equal treatment perspective implies that fairness perturbations of the input should not affect the classification label. In most cases, this assumption is valid; however, this is not always true, and even when it is, does not necessarily apply equally to all members of the group referenced in the example (e.g., not everyone who identifies as a woman can or chooses to give birth). This makes our measurements noisier. We do not currently use our perturbation method to filter out any examples from our data, because this may unintentionally lead to the over- or under-representation of certain topics and voices.

Secondly, whenever one chooses a set of demographics, they are immediately codifying particular categories and deciding which groups to include or leave out. We have chosen for now only a set of size two—gender identity and race/ethnicity. We chose these largely because they have clear and measurable effects in language (i.e., with respect to pronoun morphology, noun phrases, and names). Within these two broad groups, we have chosen to include only a subset of the possible perturbations: man-woman for gender, and Asian-Pacific Islander-Black-Hispanic-white for race/ethnicity). For the latter, this decision was driven by data availability and could reflect a US-centric bias present in the existing datasets. For the former, we intended to additionally include a gender-neutral perturbation, which could swap names that were statistically more likely to refer to men or women to those that were gender-balanced, and similarly to perturb "sister" and "brother" to "sibling", "her" and "him" to "them", and "female" and "male" to the empty string. We feel this direction is of clear importance, since it can make it possible to uncover examples that are unfair to nonbinary people. Unfortunately, there were various idiosyncrasies both in our language ("standard" American English), and in the existing datasets that made the quality of the examples decrease too drastically for us to conclude that a gender-neutral perturbation would yield useful signal. For example, for the "(fe)male" swap with the empty string, we observed that $50\%$ of the uses of these terms in MNLI were nouns; permuting them to the empty string would result in ungrammatical examples. Furthermore, in addition to case collapses between possessive forms of the pronouns (i.e., "their(s)", singular-*they* has an additional complication in that it controls plural verbal agreement, meaning that heuristic perturbations often introduce additional ungrammatical noise above and beyond the case collapses in morphological form). Due to these and other complications, we reluctantly set aside gender-neutral perturbations for future work. All this being said, there is ample room to improve upon our current fairness perturbations, not only by extending the scope of included groups, but also by exploring options for perturbation that are more flexible.

Thirdly, our fairness perturbation method might have different consequences for different tasks. For Sentiment Analysis, QA, and NLI, our spot checks suggest that data crucial for the classification is affected only rarely (see Appendix A for more discussion). However, for hate speech, the situation is a more nuanced. In data collection [71], abuse against men, straight people, white people, etc. was not considered (either as hateful or non-hateful). By performing heuristic perturbations of gender identity and race/ethnicity words, we post hoc include these sorts of examples. Since they have never seen such examples at training, this may overestimate model sensitivity to perturbation.

Finally, we would like to re-emphasize that there are many ways to measure and conceive of fairness of NLP models. We have chosen to adopt a perspective where the gender and race/ethnicity of entities in examples should not affect classification decisions. We think this is valid for the tasks hosted on Dynabench so far. This being said, models that perform well on our fairness perturbations should not be taken to necessarily be "fair" models, although we do feel it is safe to conclude that any models that perform poorly on these perturbations have room for improvement.

**Dynascore**    As discussed in Section 5, the Dynascore of a model is not fixed. Rather, it is dynamic on many levels: (i) as new models are added, the rates at which metrics are converted to units of performance may change; (ii) as the weights change, the model rankings will change. Therefore the default score is not a fixed and canonical measure of a model, and model creators should avoid reporting the Dynascore of their model out-of-context: a Dynascore only has meaning in the context of all the other models on the same leaderboard. However, users may not appreciate these caveats, and they may report the default Dynascore of their model in a research paper, much in the way they would report accuracy or F1. To discourage such behavior, the Dynaboard UI will provide clear warnings to the user, informing them of the limitations of Dynascore and the fairness and robustness metrics. If any Dynascore is reported, we recommend that users explicitly report minimally the dataset, the metric weights, and the timestamps.

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
