# A   Fairness metric

Dynabench comprises four dynamic tasks with multiple rounds of datasets that will grow over time. Given that here we have to be able to evaluate a wide variety of models, both in the loop and outside of it, we employ a black box post hoc approach, i.e., one that can be applied post-data collection to existing data, on any uploaded model, without requiring anything other than its predictions. One straightforward way to measure fairness then, is to apply clearly delimited, heuristic perturbations to existing evaluation datasets, and measure whether performance drops. Such an approach is similar to recent works that use grammars to heuristically generate pairs of examples varying in gender [58] and/or race [67] in that they utilize predefined lists of words. However, because we also want to ensure minimal consequences on our classification labels, we adopted an approach that is more targeted than grammars and also preserves the original input data distribution: we replace each word in the input data that has a clear signal about race/ethnicity and/or gender identity with a similar word referring to another group, rerun inference, and measure how many labels flipped (i.e., the difference in microaverage accuracy).

For race/ethnicity, we seeded first names across four demographic groups from a public dataset of 4,250 first names from US mortgage lending applications [69, 70]. These names cover 85.6 percent of the U.S. population, are based on the 1990 Census information on first name frequencies [70], and are licensed under a very permissive license (Creative Commons Attribution 4.0 International License). Names were associated with mutually exclusive demographic groups recognized by the 2000 and 2010 US Government Census—we selected the four groups with the most names: Asian/Pacific Islander, Black, Hispanic, white. Note: there is nothing inherently "racial" about particular names—for example, each demographic group had at least a few people named "Anna" or "Benjamin" [70]—although there are statistical trends. We selected all names for which a plurality of people of that name identified with one of the races or ethnicities, then took the 200 most frequent. We then also augmented that list with additional names popular in the literature [7, 48]. To construct our permuted test dataset, whenever we encountered a name in the input data from one of our lists, we randomly selected a name from a different race/ethnicity list and substituted it. Whenever we encountered a name in the input data which was not present in our list of names, we left it unperturbed.

For gender identity, we investigate two kinds of perturbations: names and noun phrases. For names, we affiliated the names from our race/ethnicity list with statistically likely genders, based on the U.S. Social Security Association's Lists of Baby Names (1980–2019), and performed perturbations as we did for race/ethnicity.[7] For noun phrases (i.e., pronouns and nouns), we adopted a slightly more structured approach: we still replaced words based on pair-based word lists, but we didn't do so randomly, as that could result in ungrammatical sentences (e.g., one can't replace a pronoun, like "her" with a noun like "dad" and expect no effect on the classification label). Words in the paired list that either exclusively referred to women (e.g., *her*, *sister*) or to men (e.g., *his*, *brother*) were selected by taking the union of existing popular word lists [80, 81] that had been recently extended [13, 14].[8]

Given that our perturbations are heuristic, some noise is to be expected. For example, for the names perturbations, content relevant for the classification can be affected when the name is part of a phrase referring to a known named entity. Consider "I've always enjoyed eating at *Red Robin*" being perturbed to "I've always enjoyed eating at *Red Kayla*". To mitigate this issue, we first ran an off-the-shelf named entity recognition system, and did not perturb any examples for which the system found a familiar named entity.[9] We observe that there are very few noisy examples (less than 5) resulting from NER errors. Finally, there is one irreducible type of noise arising from our heuristic approach—perturbing gender-explicit information occasionally results in unusual examples and may have consequences for the classification label: For NLI, the following hypothesis "Mothers should nurse at night" became "Fathers should nurse at night" in the context of "Failing to nurse at night can lead to painful engorgement or even breast infection". Based on spot checks performed by the authors, we conclude that noise resulting from explicit gender information is also rare (only three out of 122 spot-checked sentences). Although our approach yields enough signal to evaluate whether model performance depends on race/ethnicity and gender identity, future work exploring more flexible and adaptable approaches is encouraged.

---

[7]https://www.ssa.gov/oact/babynames/
[8]For a discussion of non-binary gender, see the Broader Impact Statement.
[9]We use the NER pipeline from spaCy [29], which was trained on OntoNotes 5.

## B  Robustness metric

For the robustness evaluation, we use a post hoc black box approach similar to the fairness evaluation. We use TextFlint [24], an open source library for measuring model robustness that covers a wide range of text transformations. We apply a family of universal transformations from TextFlint, namely `Contraction`, `Keyboard`, `Ocr`, `Punctuation`, `SpellingError`, `Typos` and `WordCase`, as we focus on typographical errors for our robustness perturbations.

The robustness metric is computed as the percentage of unchanged predictions before and after perturbation. The assumption is that a robust model should not change its predictions upon such perturbations in input. That is to say, a model is deemed more robust if it has a higher robustness metric (i.e., a lower difference between original and perturbed examples).

## C  Model Details

We selected and trained a diverse set of models to demonstrate the value of the platform, as well as to provide a sense of the current state of the art across our multiple metrics. We did not tune hyperparameters, instead using default training hyperparameters.

**Training Data**  The NLI models are trained on ANLI [49] combined with MNLI [74], where ANLI is upsampled by a ratio of 2 to balance the data. The QA models are trained on SQuAD [57] combined with Adversarial QA [1]. The hate speech models are trained on rounds 0 through 4 of the Learning from the Worst dataset [71], with an upsampling ratio for each of 1, 5, 100, 1, and 1, respectively. The sentiment models are trained on of Dynasent [55], where round 2 was upsampled by a ratio of 3. Learning from the Worst [71], provides the upsample ratios used for hate speech, and we also performed early stopping for the hate speech transformers with the round 4 dev set. The rest of the upsampling was done to give the Dynabench data more influence in the training routine, as the leaderboard test sets on our evaluation platform are mostly comprised of Dynabench data.

**Transformers**  All of the transformer models are the base versions provided in the HuggingFace transformers library [76], except ELECTRA which is the large discriminator version. ELECTRA-large was used to test the capacity of our framework to handle larger transformer models.

We used the default training hyperparameters from HuggingFace's transformers training scripts.

The T5 models were trained with early stopping on SQuAD + Adversarial QA for QA, and the last round of the Dynabench data for the other tasks. For 2-way sequence classification, the model was trained to predict the first token of one of the strings "0", "1". For 3-way sequence classification, it was trained to predict the first token of one of the strings "0", "1", "2". There is no universal default way to handle which token to output from the T5 for sequence classification, as every sequence classification task is different. We used the training hyperparameters from the HuggingFace community T5 for text classification example.[10] An exception is that we trained for 3 epochs for text classification to match the BERT-style model default (the default for T5 is 2).

**FastText**  All of the FastText models were the text classification version trained with an initial learning rate of 1, 25 epochs, 2-grams, bucket size of 200000, 50 dimensions, and hierarchical SoftMax. This is the default, for large datasets, that is listed on FastText's website.[11]

**BiDAF**  BiDAF was trained with the default settings from the AllenNLP [20] BiDAF trainer.[12]

**Majority Baseline**  The majority label was determined from the corresponding train set. For example, the MNLI train set has frequency-matched labels, and the ANLI train set has entailed, neutral, contradictory splits of 52,111 / 68,789 / 41,965. So the majority baseline for NLI was to always return neutral.

---

[10]See https://huggingface.co/transformers/community.html and https://github.com/patil-suraj/exploring-T5.

[11]https://fasttext.cc/docs/en/supervised-tutorial.html

[12]https://github.com/allenai/allennlp

## D Screenshots

Please also see the screen recording provided in the supplementary material to get a better sense of how the interface works.

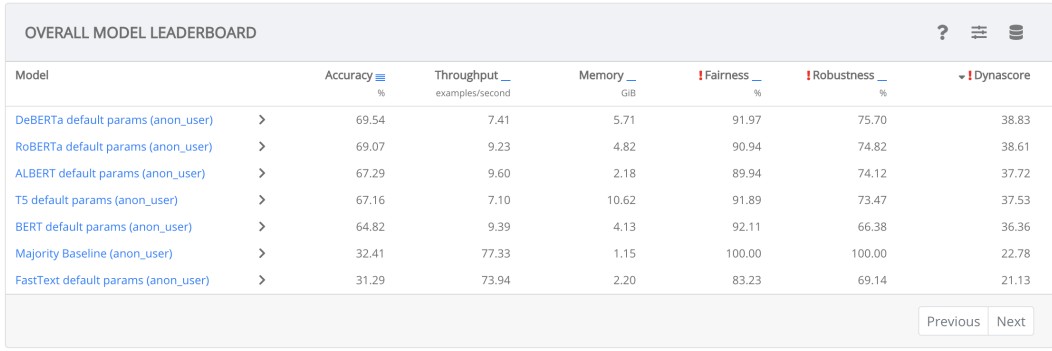

Figure 2: NLI Dynaboard example.

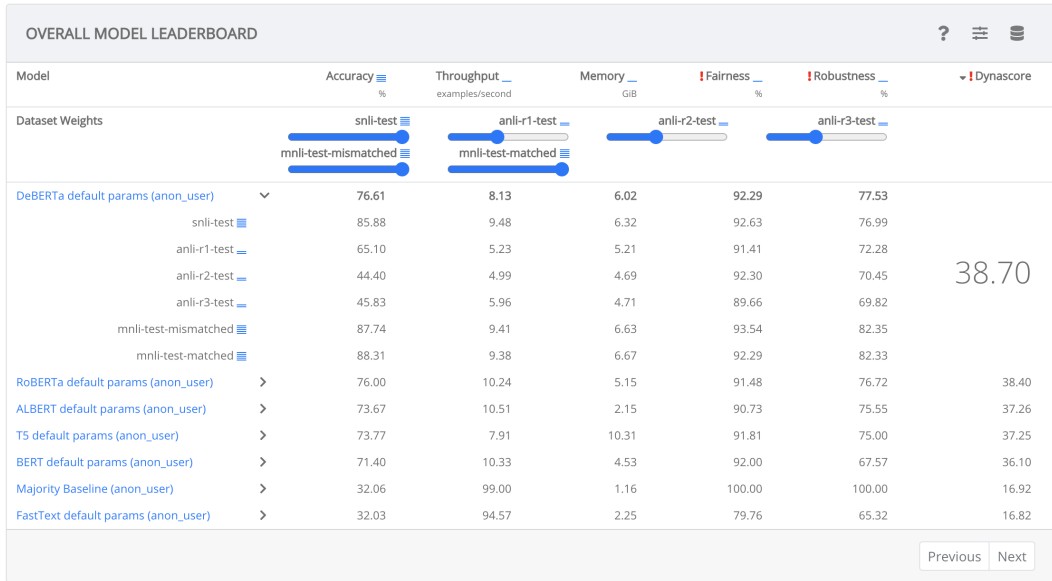

Figure 3: NLI Dynaboard example, with dataset weight sliders and finer-grained metrics displayed.

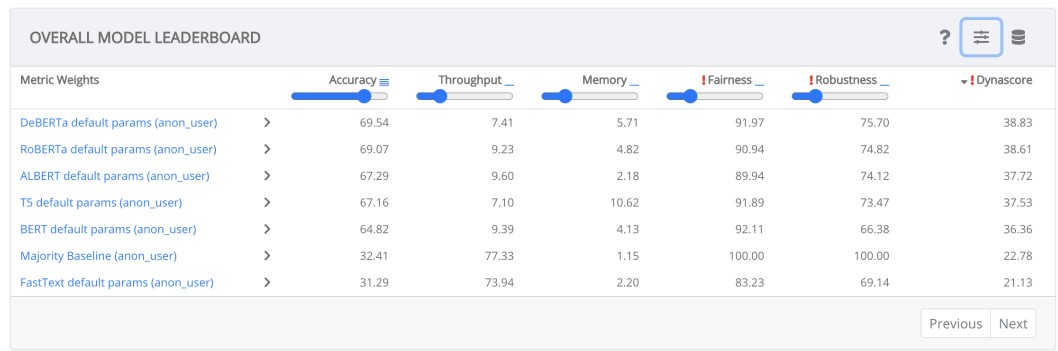

Figure 4: NLI Dynaboard example, with metric weight sliders in default configuration.

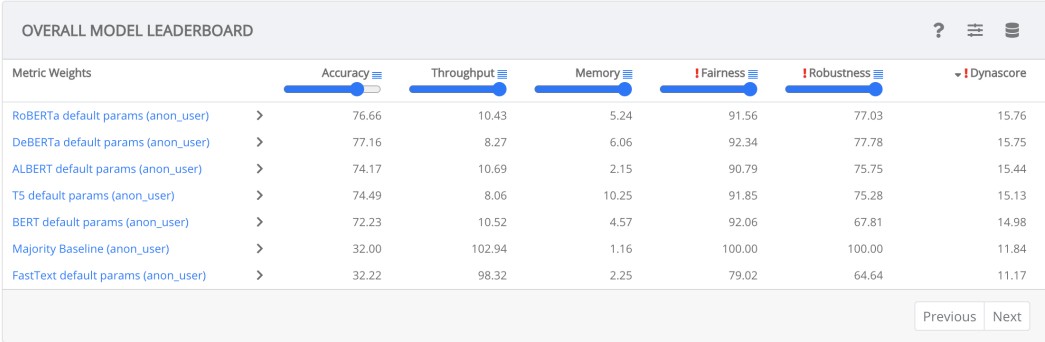

Figure 5: NLI Dynaboard example, with both dataset weights and metric weight sliders modified.

Leaderboard Datasets

| | |
|---|---|
| snli-test | 85.88 |
| mnli-test-mismatched | 87.74 |
| mnli-test-matched | 88.31 |
| anli-r1-test | 65.10 |
| anli-r2-test | 44.40 |
| anli-r3-test | 45.83 |

Non-Leaderboard Datasets

| | | |
|---|---|---|
| superglue-winogender | | 59.55 |
| mnli-dev-mismatched | | 88.27 |
| mnli-dev-matched | | 89.13 |
| snli-dev | | 85.49 |
| hans | > | 73.68 |
| nli-stress-test | > | 77.56 |
| anli-r1-dev | | 63.90 |
| anli-r2-dev | | 46.20 |
| anli-r3-dev | | 45.42 |

Figure 6: DeBERTa on NLI example, part of the model page, showing non-leaderboard evaluation-as-a-service dataset results as well.