# OpenReview forum: "Dynaboard: An Evaluation-As-A-Service Platform for Holistic Next-Generation Benchmarking"
_NeurIPS.cc/2021/Conference — NeurIPS 2021 Poster_

### Official Review · Reviewer_YZFF · 2021-07-15

**Rating:** 5
**Confidence:** 4

**Summary:**

The paper proposes an evaluation platform "DynaBoard" to automatic evaluate the models for different NLP tasks according to 5 metrics: performance, throughput, memory, fairness, and robustness. The scores from these metrics can be aggregated according to the custom preferences of each user. The models can then be ranked according to the aggregated score.

**Limitations And Societal Impact:**

Yes

**Main Review:**

**Pros**

1. The initiative or rather the attempt to provide a holistic evaluation score of the models will be very useful to the community and especially to people working in the application of NLP algorithms to the end tasks.

2. A uniform evaluation of the approaches would make the results of the papers more trustable and designing experiments more reproducible.


**Cons**:

Major ones:
* Most research codes are just written to obtain the maximum performance on a task and are not optimized for throughput and memory. Due to this, the Dynascore would reflect the memory usage and throughput of an unoptimized implementation and this would make the score more implementation dependent rather than method dependent.

* It is mentioned that currently the evaluation is done on CPUs. However, this is not scalable for larger models which are often more accurate. In addition, often the test data can be quite long and evaluating the model on CPU would not reflect the true throughput and memory usage if the same evaluation was carried out on GPUs.

Minor ones:
* Currently, the authors select just the pre-trained language models for evaluation. Models specialized to tasks should also be included as part of evaluation.

* Due to the paper's emphasis on benchmarking different aspects of the models performance, I feel this paper is more suited in the Datasets and Benchmarks Tracks of NeurIPS rather than in the main conference track. (https://openreview.net/group?id=NeurIPS.cc/2021/Track/Datasets_and_Benchmarks/Round1)


**Questions for the authors**

1. Do you have plans to include more NLP tasks such as machine translation, open-domain retrieval etc? If so, how will you ensure the fairness and robustness metrics are consistently improved for all the tasks?

2. How do you plan to keep the platform running in a sustainable manner in the future? The growth of models and increase in tasks would require an increasing amount of compute to be spent in evaluation and storage in hosting the models.

Depending on the authors response to the above points, I am willing to increase my scores.

**Time Spent Reviewing:**

5 hours

---

> ### Author Response · Authors · 2021-08-09
> **Response to Reviewer YZFF**
>
> Thank you for your thoughtful comments. We are happy that you think our contribution will be very useful to the community and that it will make papers more trustable and reproducible. To address your concerns:
>
> 1. We agree that research code may not always be optimized for throughput and memory (yet). However, we argue that our approach is a way to incentivize the community to think more deeply about important things like environmental footprint and accessibility of models to people blessed with fewer computational resources. For tasks or projects where optimizing for throughput or memory seems premature, researchers can argue for dynascore weightings that reflect that. This kind of flexibility in the face of differing priorities is indeed a strength of our overall approach.
> 2. At submission, we did not support GPUs yet, but you are absolutely right that scaling large models would be problematic on CPU only. Hence, we have now added GPU support, and have successfully utilized this in hosting the large-scale multilingual machine translation shared task at WMT.
> 3. We appreciate your suggestion of adding more task-specific models. In this case, we focused on models that were included in the original papers for the given task and followed prior work in this space. We note that we did include BiDAF, which is specific to Question Answering. We would be happy to also add e.g. InferSent for NLI, as well as other task-specific models (suggestions welcome). The hope is of course that the community will add more models over time as well!
>
> To answer your questions:
>
> 1. We would love to add tasks like MT and retrieval. In fact, as mentioned above we have already successfully hosted a WMT shared task on the platform. In that case, we did not yet include fairness and robustness metrics and focused solely on accuracy, with plans to incorporate additional metrics later. There is a lot of interesting work on fairness in MT that would be interesting to incorporate, see e.g. this blog post from Google: https://ai.googleblog.com/2021/06/a-dataset-for-studying-gender-bias-in.html, but a lot of work needs to be done to flesh this out in more detail.
> 2. A good example that it is possible to run a platform like this in the long term is MLPerf, which is owned and hosted by the MLCommons (https://mlcommons.org/en/). They have much heavier computational requirements than our platform, and are funded by industry partners. Ideally, we would find a similar solution for this platform eventually. We are currently well-funded to handle orders of magnitude more models and data than we do now.
>
> We’d like to thank you again for your very thoughtful review, and we hope that we have addressed your concerns sufficiently for your scores to be revised.

---

> > ### Author Response · Authors · 2021-08-25
> > **Has our response addressed your concerns?**
> >
> > Dear reviewer YZFF, we would be grateful if you can confirm whether our response has addressed your concerns, and let us know if any issues remain. To recap our response:
> >
> > 1. We argue that the research community, even if it doesn't always do so yet, should think about important aspects of AI model development and implementation such as greenness (i.e., throughput) and fairness, and that our approach can help incentivize this. We hope you agree.
> > 2. We have now added GPU support, and have successfully demonstrated recently that this works well on a large-scale WMT task.
> >
> > Thank you again for your thoughtful review. Let us know if you have any other questions!

---

> > > ### Comment · Reviewer_YZFF · 2021-08-25
> > > **Response to author's rebuttal**
> > >
> > > Thanks for the author’s response. I have read it and decided not to change my scores. Please find the reasons below:
> > >
> > > Compared to prior benchmarking efforts which prioritized model accuracy, Dynaboard additionally aims to provide scores characterizing different model aspects of: inference throughput, inference memory usage, fairness, and robustness. However, I feel that under the current setup, apart from the common performance (accuracy) metric, the scores for the other metrics are still not reliable.
> > >
> > > **Throughput and Memory Usage**:
> > >
> > > *Careful benchmarking required on GPUs*: To report the results in Table 2, the evaluation was performed on CPUs. Given that hardware accelerators such as GPUs (and TPUs) are widely used by ML practitioners as they provide super-fast GEMM implementation that speeds up Transformer models, the scores should be reported when GPUs are used. For large models which cannot fit within one GPU, the evaluation framework should support model parallel feature. When evaluation is done using GPUs, the throughput and memory usage of different models can be better understood and thus compared against.
> > >
> > > *Concern on the usage of non-optimized implementations of the models*: As mentioned in the supplementary material, I think the paper uses model implementations from non-official and official sources. These implementations are very rarely optimized to obtain the maximum throughput or to have the minimal memory usage. This is because incorporating performance optimizations to improve the model on these aspects involves a lot of systematic study and is a different task altogether. It will not be fair to expect from researchers who come up with a new model to also provide its carefully optimized implementation. Generally, once it is demonstrated that a model performs well on a task of interest, a different team of engineers work to re-implement them in an efficient manner, which is also evident from the submissions to the MLPerf competitions.
> > >
> > > **Fairness and Robustness**
> > >
> > > As also noted by the authors, these metrics are only available only for a handful of tasks and research on them is still in its infancy. Due to this, I believe that the current set of scores would be of limited importance. However, as this also a fast evolving subfield, I expect reliable benchmarks to be available sooner or latter.
> > >
> > >
> > > Due to these reasons, I feel that Dynabench although is a work in the right direction but still is very much a work in progress and needs more work in order to be submission ready.

---

### Official Review · Reviewer_Uxmc · 2021-07-16

**Rating:** 6
**Confidence:** 3

**Summary:**

This paper introduced Dynaboard, an evaluation framework in which models are uploaded to and evaluated in the cloud in a way that is reproducible and standardized. Doing so makes it easier to track other metrics that are often neglected, such as throughput, memory usage, fairness, and robustness. Finally, the paper introduces Dynascore, a measure that aggregates performance across different metrics, the weights of which can be dynamically chosen by the user to reflect their preferences.

**Limitations And Societal Impact:**

The authors raised concerns about including a fairness metric: "Whether to include any explicit fairness metric was a difficult choice – we acknowledge that we might inadvertently facilitate false or spurious fairness claims, fairness-value hacking, or give people a false sense of fairness." This is also a concern I have, and don't think it's clear that including a bad fairness measure is better than including no fairness measure at all, so I'm not convinced of the author's choice to do so. That said, I don't think the choice is obvious either way, and think their decision is more reasonable given that it can be updated over time. Overall I'm glad they discussed this point in a reasonable amount of detail, because I think it's the sort of design decision that can matter a lot.

**Main Review:**

I am generally excited to see new approaches to benchmarking progress, and overall find the proposal of Dynaboard promising. It seems quite original (as the authors mentioned, there have been other attempts at hosting leaderboards or trying to improve reproducibility, but Dynaboard seems to cover various facets of these problems not covered by prior work), well-designed, and addresses an important problem. It would indeed be very valuable to make evaluation of models easier and more reliable, while making users more aware of the various tradeoffs different models make. I don't feel as strongly about the design of Dynascore for aggregating metrics (I would have liked more discussion about why this is particularly good and natural), but it also seems pretty reasonable.

Overall, I think Dynaboard is interesting and potentially quite important, and the paper itself is generally well-written.

Some other comments/feedback:
- One thing I like about Dynaboard is that is (to the best of my knowledge) entirely new is evaluation for users in the cloud. However, how scalable is this as models become increasingly large? For example, if in a few years we have open-source models larger than GPT-3, which would presumably require a huge memory and computational footprint, will you still be able to host this evaluation (for free, I'm assuming)?
- Moreover, there are lots of additional metrics and datasets that I would like to see added, I think it would be stronger if many more were already added. This isn't a dealbreaker because it's deliberately extensible, but this may be important for Dynaboard to gain traction, and it makes the framework more exciting to me because of what it could become in the future than because of what it is currently.

------------
Update: I have read the authors' response and will keep my score at a 6.

**Time Spent Reviewing:**

3

---

> ### Author Response · Authors · 2021-08-09
> **Response to Reviewer Uxmc**
>
> Thank you for your very helpful review! We are happy that you think our work is promising, original, well-designed and important. To address your questions:
>
> 1. Regarding scalability: this is an excellent point. We think that there is quite a lot of leeway for most types of models, beyond the absolute biggest ones. Models will scale roughly with hardware cost, so the bigger models that will exist in a few years from now may not necessarily be more expensive to host. That said, if this really becomes a problem there are also possibilities for letting the owner of the model host it in a controlled way, thus having them absorb (part of) the cost themselves.
>
> 2. Regarding additional metrics: we absolutely agree! As you said, Dynaboard is deliberately extensible and in hindsight we should have done a better job making very explicit that we hope to add many more metrics. We are not necessarily committed to any particular set, since we think that this should ultimately be up to the task owners to decide. We share your excitement for what it could become in the future!

---

### Official Review · Reviewer_mrnU · 2021-07-16

**Rating:** 7
**Confidence:** 4

**Summary:**

This paper presents a platform called Dynaboard for evaluating NLP models in a cloud server and hosting a leaderboard of submitted models. The motive of this web platform is to evaluate models on multiple metrics for performance, memory use and throughput, robustness and fairness metrics for models. Since evaluation is performed on the cloud, the platform also aims to alleviate concerns about reproducibility, accessibility and backwards/forwards compatibility. Finally, users can rank models according to an aggregated metric called Dynascore, where the aggregation criteria is determined by them. The method of computing this aggregated score is based on economic theory, representing each metric as a good, where having more of a good is more beneficial.

To compute the Dynascore measure, the authors use the idea of indifference curves, which is a set of points (models) that provide the same utility. From this curve, they compute the average marginal rate of substitution, which is the rate at which model creators are willing to tradeoff a metric for a 1-point increase in performance.

Finally, the paper presents an evaluation of multiple models on 4 NLP tasks on performance, throughput and other metrics included in Dynaboard, along with their computed Dynascore. The authors find that the rankings obtained using Dynascore are more intuitive than the average z-score and that they largely preserve the rankings on the SuperGLUE leaderboard.


**Limitations And Societal Impact:**

The authors address the limitations of this work in sec 5.4, and generally communicate that their platform is going to undergo changes over time to incorporate more acceptable definitions of different metrics. The authors also outline the societal impact of this work in the last section of the paper. Specifically, they emphasize the potential adoption of green AI through the use of this platform and highlight some issues due to the current definition of fairness within the platform.


**Main Review:**

First, the reason for building this platform is well-motivated and targets some very important problems such as reproducibility, accessibility and compatibility. Providing evaluation-as-a-service clearly has its advantages and I think this paradigm shift in evaluation is imperative. The paper is written well, provides a clear description of the motives behind this platform, and describes one method for how metrics can be aggregated using economic theory to reflect the user's needs. I did not learn a lot from the results in Table 2; perhaps the way in which these results are presented can be revisited. I do have several questions and a few concerns:

- Black box evaluation of robustness / fairness: Currently, outputting a single number for fairness and robustness makes these concepts very much like a black box when they really shouldn't be that way. It would be great if these measures could be tied to more real success/failure modes of the model, either by providing examples of where a submitted model fails at being robust / fair or documenting the harms of using a specific model.

- Incentivization: Certain research labs might not be incentivized to pay attention to some of these metrics, when they are clearly of importance to the community at large. Specifically, how does this platform incentivize researchers to still optimize for non-performance metrics? Put another way, how would this platform ensure that modeling progress is driven not just by performance improvements but also other metrics?

- Robustness / fairness definitions: The current formulations of robustness / fairness clearly have weaknesses, which the authors acknowledge in the paper. I think that if the authors envision this platform to be adapted fairly quickly, then these notions need more input from researchers in the corresponding areas. The current definitions are very limited and this might hinder users from adaption. Also, have you considered providing users the option to pick from between several options of fairness / robustness metrics? To avoid the potential harm of providing a number for fairness / robustness when it's imperfect, the authors could include detailed descriptions in the documentation of the platform (if it doesn't exist already).

- Open-sourcing: What are your plans for making this platform an open-source project? Notions of robustness, throughput etc could definitely use input from researchers in those sub-areas.

- Task ownership: How are task owners determined? Further, do you think that the task of picking the default settings for Dynascore should be democratized as well?

- Model card functionality: Have you considered incorporating the ability for users to submit model cards?

- Significance testing and multiple runs: Would it be possible to enable users to perform significance testing within the platform to compare models which have small differences along any metric? Also, do you provide standard deviations / error bars of multiple model runs with different random seeds?


**Time Spent Reviewing:**

4

---

> ### Author Response · Authors · 2021-08-09
> **Response to Reviewer mrnU**
>
> We thank you for your support, your very thoughtful comments and super helpful feedback! To address your comments:
>
> 1. Moving beyond black-box fairness and robustness: this is an excellent suggestion - we are actively working to replace these methods with finer-grained alternatives that would allow exactly what you suggest, i.e. to better document potential harms and explicitly elucidate weaknesses beyond the single number. These would be documented on the model view page, along with the other EaaS results.
> 2. Incentivizing researchers to still optimize for non-performance metrics: we hope that, by providing the right tools and making these things easy to measure, the community will start to hold everyone accountable, which will hopefully be enough of an incentive for such researchers to move beyond pure accuracy-based metrics.
> 3. Definitions: you are absolutely right, and we will definitely seek input from researchers in the corresponding areas. Yes, we are very much in favor of letting task owners (or even users, as you suggest) select for themselves what exact metrics they want to utilize. As we acknowledge in the paper, we fully expect these metrics to improve and evolve over time. We make the imperfection of these metrics very explicit in the user interface (red exclamation marks that say “Warning: this fairness metric is still under development. A high fairness score does not necessarily mean that the model is fair along all dimensions or for all definitions of fairness. See the paper for details on this metric.”)
> 4. Open sourcing: A large part of the platform (for uploading models) is already open source and we plan to open source the rest soon as well, all under a permissive license (e.g. MIT).
> 5. Task ownership: The task owners of the current tasks are recognized experts on their tasks and were asked to collaborate for that reason. In the near future, anyone who wants will be able to launch their own task on the platform. The default settings for the Dynascore are currently determined by the task owners, but we do plan to offer users the option to construct and share their own custom Dynaboard configurations via short URLs, effectively democratizing default settings, as you suggest.
> 6. Model cards: Yes, we already support this - all model uploads come with a model card template that the model owner should fill out with additional details, which are prominently displayed at the top of the model overview page.
> 7. Significance testing: This is an excellent suggestion! It would be great if we could add this. We do not currently display error bars because users only upload a single model, but what we can do is evaluate on different random subsets of the test sets to estimate confidence bounds, thanks for the suggestion!

---

> > ### Comment · Reviewer_mrnU · 2021-08-27
> > **Follow up**
> >
> > Thanks for your response!
> > I do still worry about 1. in the paper's current proposals and how an imperfect definition of robustness/fairness can potentially cause more harm than good. A platform that presents scores on these different axes is useful, as long as the authors continually monitor how users use their platform. I'd be curious to hear the authors' plans to study how Dynaboard is used over time.
> >
> > All in all, I think this is a promising direction, but I wish the work presented user studies that illustrate the usefulness of the platform, especially with regards to presenting scores on multiple metrics (I noted that the authors highlighted this as future work). For eg, in what scenarios do users weigh one metric more than the other? How do users typically adjust the weights instead of using the default ones? Even a small user study (with 10-20 participants) could have demonstrated the potential impact of this work.
> >
> > I think I'll keep my current score for now.

---

### Official Review · Reviewer_qZyj · 2021-07-18

**Rating:** 6
**Confidence:** 4

**Summary:**

The authors of this paper introduce dynaboard, an evaluation service framework that is integrated with the dynabench platform. The goal of this service is to evaluate and benchmark the progress of NLP models whilst having a focus on reproducibility and accessibility. Dyanboard allows the models to be evaluated directly in the cloud on multiple metrics, including a focus on memory usage. To compare and rank different models, the authors compute dynascore. Further, the evaluation platform provides flexibility to the end-user to customize the dynascore based on weights.

**Limitations And Societal Impact:**

Yes, the authors have acknowledged some of the limitations of their approach particularly with regards to fairness and robustness.

**Main Review:**

Strengths & Weaknesses:
1. The authors tackle an important problem of benchmarking across multiple NLP tasks by introducing a new evaluation service called dynaboard with different objectives compared to existing benchmarks that exist. Similar to existing benchmarks, dynaboard also focuses on reproducibility and accessibility that enables the models to be available. A differentiating factor compared to existing benchmarks might be the ability of dynabench to automatically evaluate a model on a new dataset or metrics.

2.  From a metrics perspective, the platforms provide the flexibility of choosing multiple task-based metrics that are aimed at not only evaluating performance but other aspects including throughput, memory, and fairness. I appreciate the authors for looking at other metrics that are focused on carbon print and memory. However, there seems to be a lack of details in terms of how they are calculated pertaining to throughput, memory. With regards to fairness, just using perturbations is not the right approach for measuring it. This may lead to false validations about the overall performance of a model. Similar concerns are applicable to the metric of robustness.

3. One of the main contributions from this work, is the calculation of the dynascore based on theory from microeconomics. From Table 2, it can be seen that the new scoring approach captures the overall performance of the system and rankings are pretty similar to SuperGLUE.  However, I feel that the main advantage of this new scoring would be reflected in its ability to capture the performance of the system over time and adjust the scoring based on new metrics.

Questions:
1. What is the level of integration between Dynabench and dynaboard? Are the models submitted to dynaboard evaluated periodically on every new test set from dynabench?
2. With regards to the design choices for the front end, is there any reasoning as to why the users were given the flexibility to pick and choose different metrics? From an evaluation consistency perspective, wouldn't it be better if all the models had some amount of default metrics (metrics for the task )and the rest of the metrics such as memory, throughput, fairness, and robustness can be optional?
3. With regards to the MRS calculation that is dependent on the performance metrics, the current equation does not take into consideration when the performance of two models are the same? Would it be better to include another parameter (Small weight) to avoid this issue?

**Time Spent Reviewing:**

30

---

> ### Author Response · Authors · 2021-08-09
> **Response to Reviewer qZyj**
>
> Thanks for your very helpful review! To address your concerns:
>
> - Details on how throughput and memory are calculated: apologies, we omitted some details due to lack of space. This information is provided in the user interface; here is the relevant text as it is currently provided (suggestions welcome): “The compute and memory metrics are computed using AWS Cloudwatch. Throughput in examples per second is computed as the total number of examples divided by the inference time in seconds. The inference time is the difference between TransformEndTime and TransformStartTime from AWS’s DescribeTransformJob API. Memory is the average of all logged MemoryUtilization data points (logged as a utilization percentage every 1 minute by AWS) during inference, which is converted into GiB by multiplying with the total available memory of the instance type. Note that this is the memory utilization of the entire model’s docker container that serves the model. Both metrics are dependent on the instance type, and contain some randomness, i.e. they are expected to change slightly every time even in exactly the same setup. In our setup, ml.m5.2xlarge is the default machine instance, which has 8 cpus and 32 GiB memory. All metrics are higher-is-better, except memory, where lower is better.”
> - Regarding fairness and robustness, in the user interface we do our best to avoid false validations by making clear that the metrics are imperfect, stating “Warning: this fairness metric is still under development. A high fairness score does not necessarily mean that the model is fair along all dimensions or for all definitions of fairness. See the paper for details on this metric”. We hope we made our reasoning about including these metrics sufficiently clear in the broader impact statement.
>
> And to answer your questions:
>
> 1. Indeed, the idea is that Dynabench and Dynaboard would be tightly coupled. For example, it would be a natural choice to put the highest scoring models on the leaderboards “in the loop” for data collection to construct new test sets, as well as to re-evaluate all models on newly collected test sets as they come out.
> 2. We argue that the evaluation consistency perspective is covered by the default weights that are set by the task owners. Users are able to adjust weights according to their own utility function, which we believe is important because different users have different needs, but the weights will always revert to the default when the user returns later. If task owners preferred, they could in fact have made the non-performance metrics completely optional in their task’s default settings.
> 3. We address this in footnote 6: since the MRS is undefined when the difference in performance between consecutive models is zero, we ignore increases in performance that fall below a predetermined threshold ε (i.e., they are not included in the AMRS calculation). As discussed in the “Threshold Sensitivity” paragraph in section 5.4, we consider ε=0.0001 to be a reasonable setting because most leaderboards only report 1 or 2 decimal points. Although changing this threshold can change model rankings, we find that, in practice, the impact of this is minimal for choices in [0.0001, 0.01]. Although it is possible to make this threshold variable, at the moment we fix it at a conservative value to avoid confusing the user.

---

> > ### Comment · Reviewer_qZyj · 2021-08-22
> > **Further Questions**
> >
> > Thanks for your detailed answer about memory and throughput. It does answer the question I had previously raised. With regards to dynabench, there is an emphasis on dynamic data collection to prevent overfitting on static benchmarks, With the tight coupling between dynabench and dynaboard:
> > (1) would the submitted models be regularly evaluated on newly collected test benchmarks or just on the benchmark available at the time of submission?
> > (2) If no to prior question, how would models be compared when they are evaluated on different test sets?

---

> > > ### Author Response · Authors · 2021-08-23
> > > **Answer to further questions**
> > >
> > > Thanks! We're happy that our reply addressed your concerns. Regarding your further question: yes, the idea is that submitted models would be regularly evaluated on newly collected test benchmarks, or even with humans in the loop directly! One of the major features of our platform is that we have backward compatibility, and can (re-)evaluate all models on all test sets, ensuring that models can be compared properly and fairly. Does that answer your question?

---

> > > > ### Author Response · Authors · 2021-08-28
> > > > **Follow-up**
> > > >
> > > > Hi! Just checking in here - thank you again for your thoughtful questions. If our response addressed your concerns, that's great! If not, we are happy to hear more.

---

> > > > > ### Comment · Reviewer_qZyj · 2021-08-30
> > > > > **Follow up**
> > > > >
> > > > > I have no further questions. Thanks for answering all my questions. I have taken the author's response into consideration along with the response to other reviews. I applaud the authors for taking fairness and robustness into consideration as a part of the benchmark,. However, I feel that this seems to be in the preliminary stages and would prefer more analysis from the perspective of fairness and robustness. I would like to stick to my current score

---

### Decision · Program_Chairs · 2021-09-27

**Decision:**

Accept (Poster)

**Comment:**

This work on the Dynaboard benchmark is an important contribution to the community as it touches several important points: (a) evaluate a model more throughly not just accuracy (incentivizing aspects such as greenness and fairness) and (b) make evaluation easier, more accurate, and even forward compatible. I am somewhat torn about the decision whether to accept this work to the main conference. While I share some of the concerns from reviewer YZFF and think that there is little technical "machine learning" content (there's an interesting technical content on computing Dynascore, but it's not really about ML), I recommend Accept with the hope that the paper will spur further progress on evaluation in our field.